# Type I Cystatin Derived from *Fasciola gigantica* Suppresses Macrophage-Mediated Inflammatory Responses

**DOI:** 10.3390/pathogens12030395

**Published:** 2023-03-01

**Authors:** Pathanin Chantree, Mayuri Tarasuk, Parisa Prathaphan, Jittiporn Ruangtong, Mantana Jamklang, Sirilak Chumkiew, Pongsakorn Martviset

**Affiliations:** 1Department of Preclinical Science, Faculty of Medicine, Thammasat University, Pathumthani 12120, Thailand; 2Thammasat University Research Unit in Nutraceuticals and Food Safety, Thammasat University, Pathumthani 12120, Thailand; 3Research Group in Medical Biomolecules, Faculty of Medicine, Thammasat University, Pathumthani 12120, Thailand; 4Graduate Program in Bioclinical Sciences, Chulabhorn International College of Medicine, Thammasat University, Pathumthani 12120, Thailand; 5Institute of Science, Suranaree University of Technology, Nakhon Ratchasima 30000, Thailand

**Keywords:** stefin-1, liver fluke, anti-inflammation, THP-1, RAW 264.7, NF-κB

## Abstract

There is an inverse relationship between the high incidence of helminth infection and the low incidence of inflammatory disease. Hence, it may be that helminth molecules have anti-inflammatory effects. Helminth cystatins are being extensively studied for anti-inflammatory potential. Therefore, in this study, the recombinant type I cystatin (stefin-1) of *Fasciola gigantica* (rFgCyst) was verified to have LPS-activated anti-inflammatory potential, including in human THP-1-derived macrophages and RAW 264.7 murine macrophages. The results from the MTT assay suggest that rFgCyst did not alter cell viability; moreover, it exerted anti-inflammatory activity by decreasing the production of proinflammatory cytokines and mediators, including IL-1β, IL-6, IL-8, TNF-α, iNOS, and COX-2 at the gene transcription and protein expression levels, as determined by qRT-PCR and Western blot analysis, respectively. Further, the secretion levels of IL-1β, IL-6, and TNF-α determined by ELISA and the NO production level determined by the Griess test were decreased. Furthermore, in Western blot analysis, the anti-inflammatory effects involved the downregulation of pIKKα/β, pIκBα, and pNF-κB in the NF-κB signaling pathway, hence reducing the translocation from the cytosol into the nucleus of pNF-κB, which subsequently turned on the gene of proinflammatory molecules. Therefore, cystatin type 1 of *F. gigantica* is a potential candidate for inflammatory disease treatment.

## 1. Introduction

Inflammation is a protective mechanism of the body to eliminate noxious stimuli, for example, mechanical damage, toxins, and especially infection by pathogens [1]. However, the typical inflammatory response mechanisms occur differently for harmful stimuli. These processes include recognizing harmful stimuli by cell surface pattern receptors, activating inflammatory pathways, releasing inflammatory-related markers, and recruiting inflammatory cells [2]. Unfortunately, the extreme production of those molecules may cause negative effects such as inflammatory-related diseases [3,4]. During the inflammation response, macrophages are crucial immune cells that play a pivotal role in the inflammatory process by releasing several proinflammatory cytokines, such as interleukin-1β (IL-1β), IL-6, and tumor necrosis factor-alpha (TNF-α) as well as proinflammatory mediators such as prostaglandin E2 (PGE2) and nitric oxide (NO), which are enzymatically synthesized by cyclooxygenase-2 (COX-2) and inducible nitric oxide synthase (iNOS), respectively [5,6,7,8]. Hence, patients with inflammatory-related diseases require strategies to control excessive inflammation for effective treatment.

Regarding the relationship between chronic inflammation-associated diseases (CIADs) and parasitic infections, CIADs were often observed in developed countries with good hygiene and low cases of parasite infection, while fewer cases of CIADs occurred in developing countries where high cases of parasitic infection existed [9,10]. The inverse association between the incidence of CIADs and parasitic infection, especially by helminths, was elucidated [11,12]. Furthermore, moderate helminth infection was found to reduce symptoms in CIADs [9]. The reasons for this relationship may involve helminths’ activities inside the host body, including evasion from the host immune system. They can regulate host immune responses by producing and secreting several immunomodulatory molecules [13] found in their excretory/secretory products (ESPs) [14,15]. Due to immunomodulatory properties, it has been suggested that ESPs from helminths could probably be considered therapeutic agents for inflammatory-related diseases such as autoimmune, metabolic disorders, and allergies [16,17,18]. The suppression of the inflammation response may involve the inhibition of macrophage activation [19]. Hence, these helminth molecules should be further studied for the development of CIAD treatments.

Cystatin is a cysteine protease inhibitor that has been successfully characterized in various organisms and different cystatins are produced in almost all nucleated cells of humans and helminths [20]. Currently, helminth cystatins have been widely studied as an immunomodulatory molecule involved in the evasion of host immune responses [20]. Regarding its amino acid sequence and structure variation, there are three subgroups of cystatin. Type I cystatins (stefins) including cystatins A and B are unglycosylated proteins without disulfide bonds with nearly 100 amino acids. They are usually found as intracellular proteins but are occasionally secreted. Type II cystatins comprise 120 amino acids with two conserved disulfide bonds to the carboxyl terminus. They are mostly extracellular and secretory proteins, including cystatins C, D, E/M, F, G, S, SA, and SN. Lastly, type III cystatins or kininogens comprise around 350 amino acids. In humans, they are synthesized in the liver as multidomain glycoproteins and are mostly found in plasma and involved in both innate and adaptive immune responses [20,21]. 

For an example of the anti-inflammatory effect of helminths’ cystatins, the type II cystatin of *Schistosoma japonicum* (Sj-Cys) could diminish the production of IL-6 and TNF-α in a lipopolysaccharide (LPS)-induced murine macrophage cell line (RAW264.7 cells) [22]. Furthermore, a previous study demonstrated that type I Sj-Cys decreased the proinflammatory cytokines that cause reduced inflammation and multiple organ failure induced by sepsis [23]. Treatment with type II cystatin of *Litomosoides sigmodontis* altered the production of iNOS, the mediated molecule for nitric oxide (NO) production, during the infection of microfilaria [24]. Type II cystatin of *Acanthocheilonema viteae,* a filarial worm, was able to inhibit T-cell proliferation and stimulate the secretion of IL-10, which is an anti-inflammatory cytokine [25]. Interestingly, the recombinant type II cystatin of *Trichinella spiralis* inhibits the release of proinflammatory cytokines and mediators including IL-1β, IFN-γ, TNF-α, and iNOS, and induces M2 polarization with alteration of the antigen processing by downregulating MHC class II expression (MHC-II) in mouse bone marrow-derived macrophages (BMDMs) under activation of LPS [26,27].

Among the parasitic helminths, *Fasciola* spp. or liver fluke is a foodborne parasite that causes fascioliasis, which is an important zoonotic disease that mainly infects livestock and decreases the growth rate as well as the amount and quality of livestock production [28]. The current economic loss in animal production due to fascioliasis is estimated to be 3.2 billion US dollars [29]. Importantly, it is evaluated that 2.4 million people are infected, and 180 million people are at risk in over 70 countries [30]. The host is infected by consuming contaminated food such as water plants with metacercariae, which is the infective stage [31]. The newly excysted juveniles (NEJ) emerge from the metacercariae stage in the host’s intestine and migrate to the liver by rapidly passing through the gut wall. Moreover, Fasciola migration also frequently occurs in other organs such as the lung and even the eyes [32]. During migration, NEJ can secrete ESPs composed of cystatins and cathepsin B and L cysteine proteases for tissue invasion and host immune response evasion [33,34]. The ESPs of *Fasciola hepatica* can inhibit the activity of host immune responses by inducing the apoptosis of immune cells and host immune molecule degradation [35,36,37]. Previous studies have shown that *F. hepatica* stefins-1, stefins-2, and stefins-3 were found in *F. hepatica* ESPs and could inhibit host cathepsin [38]. Moreover, the recombinant *F. hepatica* cystatin (rFhcystatin) inhibited the expression of cellular NO, IL-6, and TNF-α, and promoted the expression of IL-10 and transforming growth factor-β in LPS-activated macrophages [39]. 

For *Fasciola gigantica*, a previous study suggested that type I cystatin is a major component of its ESPs. The recombinant *F. gigantica* type I cystatin (FgCystatin) showed the inhibition of cysteine protease activity, which may affect the antigen-presenting process of antigen-presenting cells [40]. Nevertheless, the roles of FgCystatin in the host’s immune responses have not been elucidated in detail. Hence, this study aims to evaluate the anti-inflammatory properties of recombinant type 1 FgCystatin (rFgCyst) in LPS-activated macrophages. The results will provide important knowledge for developing FgCystatin to be used as a potential anti-inflammatory molecule in the future. 

## 2. Materials and Methods

### 2.1. Production of Recombinant FgCyst (rFgCyst) and Endotoxin Removal 

The nucleotide sequence encoding *F. gigantica* type I cystatin (FgCyst) was obtained from the Genbank database (accession no. FJ827152). The recombinant FgCyst (rFgCyst) was produced in an *E. coli* expression system, as previously described [40,41]. Adult *F. gigantica* were collected from naturally infected cattle sacrificed at the local slaughterhouses in Pathumthani province, Thailand. The worms were washed several times with 0.85% NaCl to remove debris and excessive bile contents before being used. The total RNA of adult *F. gigantica* was isolated using TRIzol™ reagent (Invitrogen™, Thermo Fisher Scientific, Carlsbad, CA, USA) and the concentration was measured using NanoDrop™ 2000/2000c Spectrophotometers (Thermo Scientific™, Wilmington, DE, USA). Five hundred nanograms of DNase I-treated total RNA was used to synthesize the first-strand cDNA by oligo-d(T)18 primer using a RevertAid First-Strand cDNA Synthesis Kit (Thermo Fisher Scientific, Vilnius, Lithuania). The cDNA quantity and quantity were measured using NanoDrop™ 2000/2000c Spectrophotometers (Thermo Scientific™, Wilmington, DE, USA). One hundred nanograms of cDNA were used as a template to amplify FgCyst fragments by PCR using a specific primer as follows: forward primer 5′-GGA TCC ATG ATG TGC GGC GGC TGC-3′ and reverse primer 5′-CTG CAG GAG TACCGA TCA TGA TC-3′, which incorporated *Bam*HI and *Pst*I recognition sites. The PCR product was purified and cloned into a pGEM-T easy vector (Promega, Madison, WI, USA). The recombinant plasmid, pGEM-T easy containing FgCyst, was isolated by using a QIAGEN Plasmid Mini Kit (Qiagen, Hilden, Germany), and the plasmid at a concentration of 0.5–1 μg/μL was used to confirm the correction of the nucleotide sequence by DNA sequencing (Solgent Co., Ltd., Daejeon, Korea) (Appendix A). FgCyst fragments were then subcloned into a pQE-30 expression vector (Qiagen, Hilden, Germany) that contained an N-terminus histidine tag, then transformed into M15 *E. coli*. For rFgCyst production, the positive transformant of M15/pQE-30 containing FgCyst was cultured until the OD_600_ reached 0.4–0.6 before being induced with 1 mM isopropyl β-d-1-thiogalactopyranoside (IPTG) at 37 °C for 2 h. Bacterial cells were harvested and rFgCyst was purified under native conditions using high-performance Ni Sepharose^®^ (Cytiva, Uppsala, Sweden) [42]. 

The contaminating endotoxin was removed by phase separation method using Triton X-114 (Sigma Aldrich, Darmstadt, Germany) for 20–30 rounds [43,44]. Afterward, the level of endotoxin was measured using a ToxinSensor™ Chromogenic LAL Endotoxin Assay Kit (GenScript, Piscataway, NJ, USA). The acceptable level of contaminating endotoxin for cell culture was lower than 1 E.U./mg (<0.1 ng/mg) [45]. The endotoxin-free rFgCyst was dialyzed against PBS, pH 7.4, and the protein concentration was measured using a BCA protein assay (Thermo Fisher Scientific, Rockford, IL, USA). rFgCyst was verified by Western blot analysis using the anti-His tag (Bio-Rad Laboratories, Inc., Hercules, CA, USA).

### 2.2. Cell Culture and Treatments

The human leukemia monocytic cell line THP-1 was obtained from ATCC (Manassas, VA, USA). The cells were maintained in RPMI 1640 (Corning, Manassas, VA, USA) supplemented with 10% FBS, 100 U/mL antibiotic–antimycotic (Gibco, Life Technologies Corporation, Grand Island, NY, USA), 0.05 mM 2-mercaptoethanol, 1 mM sodium pyruvate, 2 mM L-glutamine, 10 mM HEPES, and 2.5 g/L glucose, at 37 °C in a 5% CO_2_ air-humidified atmosphere.

The cells were seeded into six-well plates at 2 × 10^5^ cells/mL cell density and treated with 100 ng/mL phorbol 12-myristate 13-acetate (PMA) (Sigma Aldrich, Germany) for 48 h [46]. After differentiation, THP-1-derived macrophages were subsequently cultured in normal complete media for a further 24 h.

The murine macrophage cell line RAW 264.7 was acquired from ATCC. RAW 264.7 cells were incubated in DMEM supplemented with 10% FBS, 100 U/mL antibiotic–antimycotic (Gibco) at 37 °C in a 5% CO_2_ air-humidified atmosphere. 

### 2.3. Cell Viability Assay

The effect of rFgCyst and LPS on cell viability was determined by an MTT assay. Briefly, cells were seeded on the complete culture medium at a cell density of 1 × 10^4^ cells per well in 96-well plates. After 24 h, the media were changed for a medium with rFgCyst in the presence or absence of 500 ng/mL LPS (Sigma Aldrich, Germany). First, cells were tested only with rFgCyst concentration ranging from 5 to 50 μg/mL to exclude the cytotoxic effect of rFgCyst. Next, the experimental groups including the co-incubation of rFgCyst (5, 10, 20, and 50 µg/mL) or dexamethasone (Sigma Aldrich, Germany), as a positive control, were further incubated in the presence of LPS for 24 h. Thereafter, a corresponding 5 mg/mL MTT solution (Sigma, St. Loius, MO, USA) was added to each well and cultured at 37 °C for a further 3 h in the dark. After that, the supernatant was removed, and 100 μL DMSO was added to each well. The plate was placed in a plate reader and the absorbance at 490 nm was read by a Multiskan Spectrophotometer (Thermo Scientific, Rockford, IL, USA).

### 2.4. Western Blot Analysis 

The protein expression of related molecules was analyzed by Western blot following the methods of a previous study [47] with slight modification. Briefly, after treatment, a RIPA cell lysis buffer (Cell Signaling Technology^®^, Danvers, MA, USA) containing protease inhibitors (Merck Millipore Calbiochem™ Protease Inhibitor Cocktail Set III, EDTA-Free, Darmstadt, Germany) was used to extract the total cellular proteins. Thirty micrograms of the total proteins from each sample were separated on 12.5% SDS-PAGE and subsequently transferred onto nitrocellulose membranes. The nonspecific bindings were blocked using 5% BSA in pH 7.5 tris-buffered saline (TBS) for 1 h at room temperature with agitation. The membranes were then incubated with 1:1000 primary antibodies diluted in 1% BSA in TBS with Tween^®^-20 (TBST), including rabbit anti-β-actin, IL-1β, IL-6, IL-8, TNF-α, iNOS, COX-2, pIKKα/β, IκBα, pIκBα, NF-κB, and pNF-κB (Cell Signaling Technology^®^) at 4 °C overnight with agitation. Thereafter, the membranes were washed three times in TBST and further incubated with 1:15,000 goat anti-rabbit IgG (H + L) secondary antibody, AP (Life Technologies, Carlsbad, CA, USA) diluted in 0.01 M TBST for 1 h at room temperature. The corresponding targeted proteins were visualized using 1-Step™ NBT/BCIP Substrate Solution (Thermo Scientific). Protein bands’ intensities were quantified using ImageJ software version 1.53t (available at https://imagej.nih.gov/ij/download.html; accessed on 24 August 2022).

### 2.5. Reverse Transcription Real-Time PCR

The gene expressions of IL-1β, IL-6, IL-8, TNF-α, iNOS, COX-2, and GAPDH (a housekeeping control), were evaluated. The total RNA of cells was extracted using TRIzol reagent (Invitrogen, Carlsbad, CA, USA), and first-strand cDNA from purified total RNA was synthesized using the SuperScript™ III First-Strand Synthesis System (Thermo Scientific) according to the manufacturer’s instructions. The mRNA levels were analyzed using quantitative real-time PCR and iTaq Universal SYBR Green Supermix (Bio-Rad Laboratories). The analysis was carried out on a StepOne™ Real-Time PCR System (Applied Biosystems, Foster City, CA, USA). Data were collected and analyzed using the 2^−ΔΔCT^ relative quantification method [48]. The values are shown as the fold change relative to the control. The primer sequences used for the real-time PCR are listed in Table 1.

### 2.6. Evaluation of Cytokine Secretion by ELISA Assay

To determine the secretion levels of major proinflammatory cytokines including IL-1β, IL-6, and TNF-α, the cell-free culture media of both cells were used to quantify these cytokines by using sandwich ELISA kits (Sigma Aldrich, Merck KGaA, Germany) according to the manufacturer’s procedure. Briefly, all reagents and samples were brought to room temperature (18–25 °C) before use; 100 µL of each standard and sample were added to the appropriate wells and incubated for 2.5 h at room temperature with gentle shaking. After that, the solutions were discarded and washed four times with 1X wash solution. Next, 100 µL of 1x detection antibody was added to each well and incubated for 1 h at room temperature with gentle shaking. After the washing step, 100 µL of prepared streptavidin was added to each well and incubated for 45 min at room temperature with gentle shaking. The TMB one-step substrate reagent was added to each well and incubated for 30 min at room temperature in the dark with gentle shaking. Then, 50 µL of stop solution was added to each well. The absorbance was read at 450 nm using a Multiskan Spectrophotometer (Thermo Scientific)

### 2.7. Evaluation of NO by Griess Test

The level of NO secretion was determined in the culture media using a Griess reaction [49]. The cell-free culture media were collected after treatment, then 50 µL of culture media were co-incubated with an equal volume of modified Griess reagent (Sigma Aldrich, St. Louis, MO, USA) for 10 min at room temperature. Absorbance was measured at 540 nm using a Multiskan Spectrophotometer (Thermo Scientific). The nitrite concentration of each sample was evaluated using a standard curve of prepared sodium nitrite solution.

### 2.8. Statistical Analysis

All data were presented as the mean ± standard deviation (SD). One-way analysis of variance (ANOVA) was used to determine the differences between quantitative values, followed by Dunnett’s multiple comparison test. The statistical calculations and data illustrations were performed using GraphPad Prism software version 9.3.1 (GraphPad Software, San Diego, CA, USA). A *p*-value < 0.05 was considered statistically significant. Three independent experiments were performed in triplicate.

## 3. Results

### 3.1. Cytotoxicity of rFgCyst

rFgCyst was successfully produced in an Escherichia coli (*E. coli*) expression system using IPTG induction for 2 h (Appendix A). rFgCyst was natively purified using Ni Sepharose affinity chromatography, which excessively eluted in E1–E2 fractions (Appendix A). The purified rFgCyst demonstrated two major bands at 11 kDa and approximately 22 kDa, indicating a dimer, as previously described [40]. The purified, endotoxin-free, and dialyzed rFgCyst were illustrated in the same pattern (Appendix A). The Western blot analysis using the anti-His tag demonstrated the corresponding production of rFgCyst together with a histidine tag, as shown in Appendix A.

For the optimization of the rFgCyst concentrations used for other experiments, cell cytotoxicity was determined. rFgCyst was added to the culture media in final concentrations ranging from 5 to 50 µg/mL. The cell viabilities of RAW264.7 and human leukemia monocytic cell line (THP-1)-derived macrophages were evaluated using the 1-(4,5-dimethylthiazol-2-yl)-3,5-diphenylformazan (MTT) assay to sequester any harmful effect of rFgCyst. The results of this assay showed that the treatment of RAW264.7 and THP-1-derived macrophages with various concentrations of rFgCyst for 24 h did not alter cell viability (Figure 1A).

In the experimental groups that were treated with rFgCyst, LPS, and the co-incubation of rFgCyst (5, 10, 20, and 50 µg/mL) or dexamethasone, a standard drug, with LPS for 24 h, the cell viability values of all treated groups were not significantly different from those of the control group (Figure 1B). Therefore, rFgCyst at concentrations of 5, 10, and 20 µg/mL were safely used for further experiments.

### 3.2. Inhibitory Effect of rFgCyst on LPS-Induced Inflammatory mRNA Expression

In order to determine the LPS-induced release of proinflammatory cytokines including IL-1β, IL-6, IL-8, and TNF-α and inflammatory-related molecules, including COX-2 and iNOS, we evaluated the mRNA expression of the abovementioned molecules by qRT-PCR. As shown in Figure 2A,B for THP-1-derived macrophages and Figure 2C,D for the RAW264.7 cell, LPS stimulation of THP-1-derived macrophages and RAW264.7 cells resulted in a drastic increase in the mRNA levels of IL-1β (*p* < 0.0001; Figure 2A,C), IL-6 (*p* < 0.0001; Figure 2A,C), IL-8 (*p* < 0.0001; Figure 2A), and TNF-α (*p* < 0.0001; Figure 2B,D), along with an increase in those of COX-2 (*p* < 0.0001; Figure 2B,D) and iNOS (*p* < 0.0001; Figure 2B,D). 

The co-incubation of rFgCyst with LPS in THP-1-derived macrophages and RAW264.7 cells strongly decreased the mRNA expression of these inflammatory molecules in a concentration-dependent manner (Figure 2). The IL-6 mRNA increase was almost 30- and 20-fold higher than that found in THP-1-derived macrophages and RAW264.7 control cells, respectively (Figure 2A,C). This effect was counteracted by co-incubation with rFgCyst and dexamethasone (*p* < 0.001; Figure 2A,C). Furthermore, the LPS-induced increases in IL-1β (~20-fold higher than controls), IL-8 (~20-fold higher than controls), TNF-α (~30-fold higher than controls), COX-2 (~20-fold higher than controls), and iNOS (~20–30-fold higher than controls) gene expression were reduced by co-incubation with rFgCyst and dexamethasone with a range of statistical significance (Figure 2).

### 3.3. Inhibitory Effect of rFgCyst on the Expression of Proinflammatory Cytokines in LPS-Stimulated Macrophages

To assess whether rFgCyst inhibits the increase in proinflammatory cytokines and related mediators, the protein expressions of these molecules were determined by Western blot analysis. It was found to correspond with the trend of mRNA expression: the relative expression levels of proinflammatory cytokines and related mediators massively increased following LPS treatment in THP-1-derived macrophages, as shown in Figure 3A–C, and RAW264.7 cells, as shown in Figure 3D–F. On the contrary, the co-incubation of rFgCyst with LPS in THP-1-derived macrophages and RAW264.7 cells decreased the protein expression of these inflammatory molecules in a concentration-dependent manner. However, groups treated with 5 µg/mL revealed either a few significant decreases in IL-6 levels in both THP-1-derived macrophages (Figure 3B) and RAW264.7 cells (Figure 3E) or no significant decrease in IL-1β, IL-8 (Figure 3B), and TNF-α (Figure 3C) of THP-1-derived macrophages and IL-1 β (Figure 3E) and TNF-α (Figure 3E) of RAW264.7 cells.

### 3.4. Inhibitory Effect of rFgCyst on the Production of Soluble Inflammatory Cytokines in LPS-Stimulated Macrophages

To confirm the effect of rFgCyst on LPS-stimulated IL-1β, IL-6, and TNF-α production and release, the culture media from the THP-1-derived macrophages and RAW264.7 cells were collected at the end of the experiment. The supernatants from each group were measured for the level of respective cytokines using sandwich ELISA kits. As shown in Figure 4, the treatment of THP-1-derived macrophages (Figure 4A) and RAW264.7 cells (Figure 4B) with LPS alone for 24 h resulted in a massive release of IL-1β (*p* < 0.0001), IL-6 (*p* < 0.0001), and TNF-α (*p* < 0.0001) compared to control groups. The co-incubation with rFgCyst reduced the overproduction of IL-1β, IL-6, and TNF-α proinflammatory cytokines in LPS-activated THP-1-derived macrophages and RAW264.7 cells in a concentration-dependent manner, but again, groups treated with 5 µg/mL revealed either a few significant decreases in IL-6 in RAW264.7 cells (Figure 4B) or no significant decreases in IL-6 in THP-1-derived macrophages (Figure 4A) and IL-1β in RAW264.7 cells (Figure 4B).

### 3.5. Inhibitory Effect of rFgCyst on NO Production in LPS-Stimulated Macrophages

To study the anti-inflammatory activity of rFgCyst on THP-1-derived macrophages and RAW264.7 cells, the levels of the mediators NO were measured by a Griess reaction after LPS stimulation. As shown in Figure 5, LPS treatment caused a significant elevation in NO production (*p* < 0.0001) in THP-1-derived macrophages and RAW264.7 cells. This effect was significantly alleviated by co-incubation with rFgCyst in a concentration-dependent manner. The co-incubation of LPS with dexamethasone revealed the highest significant inhibition of NO release (*p* < 0.0001). 

### 3.6. Inhibitory Effect of rFgCyst on Inflammation Involves the Alteration of Nuclear Factor Kappa B (NF-κB) Signaling Pathway in LPS-Activated Macrophages

The inflammatory response is caused by the upregulation of the NF-κB signaling pathway. The final step of this pathway is the translocation of phosphorylated NF-κB (pNF-κB), a transcription factor, from the cytosol to the nucleus, which increases the expression of proinflammatory cytokines and related mediators. To determine whether rFgCyst decreases the NF-κB activation, the levels of NF-κB signaling pathway-related molecules including phosphorylated inhibitors of NF-κB kinase complex (pIKKα/β), NF-κB, pNF-κB, inhibitory IκB proteins (IκBα), and the phosphorylated inhibitory IκB proteins (pIκBα) were evaluated using Western blot analysis. The results showed that LPS treatment excessively elevated pIKKα/β, pNF-κB, and pIκBα in THP-1-derived macrophages (*p* < 0.0001; Figure 6A–C) and RAW264.7 cells (*p* < 0.0001; Figure 6D–F). Conversely, LPS treatment significantly decreased IκBα in THP-1-derived macrophages (*p* < 0.0001; Figure 6A,C) and RAW264.7 cells (*p* < 0.0001; Figure 6D,F). The co-incubation with rFgCyst significantly decreased the upregulation of pIKKα/β, pNF-κB, and pIκBα caused by LPS induction in a concentration-dependent manner in both cells (Figure 6). Interestingly, in both THP-1-derived macrophages and RAW264.7 cells, the expression level of NF-κB did not show any significant difference among groups, even in the LPS control group. 

## 4. Discussion

The activation of macrophages by bacterial lipopolysaccharide (LPS), a major pathogen-associated molecular pattern (PAMP), involves two distinct signaling pathways composed of transcription factor nuclear factor-κB (NF-κB) and the mitogen-activated protein kinase (MAPK) [50,51,52]. Meanwhile, Toll-like receptors (TRLs) such as TLR4 have a specific pattern-recognition receptor (PRR) for LPS. In the initial activation, LPS binds to lipopolysaccharide-binding protein (LPB), is then transported to the macrophage surface associated with CD14 to form a complex with MD2, and then binds with TLR4. Eventually, the formation of the TLR4-heterodimer complex binds to MyD 88 protein, thereby activating the downstream signaling pathways [53,54].

NF-κB is a transcription factor that involves immune responses, especially those involved in inflammation. This molecule normally localizes in the cytoplasm by binding with its inhibitor, IκBα [55]. After LPS binds to the TLR4 complex, the activation is triggered by the phosphorylation of the multisubunit IκB kinase (IKK) complex, which then phosphorylates IκBα, resulting in its degradation by the ubiquitin–proteasome system. Subsequently, free NF-κB dimer translocates to the nucleus, then binds to its consensus sequence on the promoter and enhancer of target genes and activates proinflammatory genes [55].

Previous studies have shown that *F. hepatica* cystatins were found in *F. hepatica* ESPs and could inhibit mammalian cathepsins and host immune cells [38]. Moreover, at the protein level, the recombinant *F. hepatica* cystatin (rFhCystatin) shared 92.2% identity with the type I cystatin of *Fasciola gigantica* and inhibited the expression of NO, IL-6, and TNF-α in LPS-activated RAW264.7 cells [39]. For *F. gigantica*, a previous study suggested that type I cystatin is a major component of ESPs and is found in both juvenile and adult tissues [40]. The 11 kDa recombinant *F. gigantica* type I cystatin (rFgCystatin) was produced and characterized for its proposed functions. By activity analysis, rFgCystatin showed inhibition for mammalian cathepsin, B, L, and S [40]. Nevertheless, the roles of FgCystatin in the host’s immune responses have not been elucidated in detail. 

In the present study, to exclude any cytotoxic effects of rFgCyst in the absence or presence of LPS, the cell viability was determined by MTT assay. The results suggested that, after 24 h of incubation in the absence of LPS, the concentration of rFgCyst, ranging from 5 to 100 µg/mL, did not alter the cell viability in THP-1-derived macrophages or RAW 264.7 cells. The same trend was shown in the co-incubation of LPS with of rFgCyst at concentrations of 5, 10, 20, and 50 µg/mL. This suggested that rFgCyst could be used safely in broad-range concentrations without any cytotoxic effects.

The anti-inflammatory effects of rFgCyst on the transcription and translation of immune-mediating proteins in THP-1-derived macrophages or RAW264.7 cells were evaluated. The mRNA and protein expression of proinflammatory cytokines including IL-1β, IL-6, IL-8, and TNF-α, and inflammatory mediators including COX-2 and iNOS were quantified using qRT-PCR and Western blot analysis, respectively. The results revealed that rFgCyst decreased the mRNA and protein expressions of these inflammatory-related molecules in a concentration-dependent manner (Figure 2 and Figure 3). To determine the secretion level, the majority of the released proinflammatory molecules including IL-1β, IL-6, TNF-α, and NO in the cultured media were evaluated. As expected, the results revealed that rFgCyst decreased the secretion of IL-1β, IL-6, TNF-α (Figure 4), and NO (Figure 5) in the culture media in both LPS-activated macrophages in a concentration-dependent manner, corresponding to a reduction in the transcription and production levels. These results indicated that rFgCyst could reduce the production of major proinflammatory cytokines and mediators in mRNA and protein with both intra- and extracellular (secretory) constituents.

IL-8 expression was examined only in THP-1-derived macrophages (Figure 3B) since rodents lack the IL-8 gene. IL-8 is a chemoattractant for neutrophils and T-lymphocytes at the site of inflammation [56,57,58]. The previous study suggested the downregulation of IL-8 by cystatin type I, stefin A, in keratinocytes stimulated with allergens [59]. For IL-1β, especially in Raw 264.7 cells (Figure 4B), the level of secretion with high statistical significance seemed to be present only at a high concentration of rFgCyst. This probably involves unique characteristics in the production and secretion of IL-1β because this cytokine is primarily produced in an inactive proprotein. Its secretion might be involved in the NOD-like receptor (NLR) in the pyrin family domain-containing protein 3 (NLRP3) inflammasome regulation [60]. Upon inflammatory stimulation, the inflammasome receptor stimulates the activation of procaspase 1 into the active form, which then cleaves pro-IL-1β, resulting in the release of active IL-1β from the cells [61]. In this regard, the effect of rFgCyst on the secretion of mature IL-1β should be investigated next to determine the mechanisms involved in the function of NLRP3 inflammasome and caspase-1 processing. 

To explain the related mechanisms resulting in the downregulation of inflammatory molecules, several molecules in the NF-κB cascade were evaluated by Western blot. As shown in Figure 6A,D for THP-1-derived macrophages and RAW 264.7 cells, respectively, pIKKα/β levels were downregulated compared to the LPS control group. The pIKKs complex is a molecule initially produced in the NF-κB signaling pathway after LPS-TRL4 binding. The activation of the TRL4 complex further induces the TRAF 6 activity to ubiquitinate TAK1 protein and then phosphorylated molecules including IKKS that further amplify downstream signaling [62]. In the tick *Dermacentor silvarum (DS)*, DsCystatin attenuated the TRL4-NF-κB signaling pathway by downregulating TRAF6 and TAK1 [63]. Furthermore, interestingly, a previous study suggested that FhCystatin could be bound to the surface of RAW 264.7 cells [39]. In this regard, rFgCyst probably controls the TRL4-NF-κB signaling pathway in the formation of the TRLs complex at the cell membrane level, which needs to be investigated further.

Thereafter, the level of phosphorylation to IκBα, which becomes phosphorylated IκBα (pIκBα), was downregulated because of the low level of pIKKα/β (Figure 6A,D); therefore, IκBα was upregulated, followed by a decrease in phosphorylation activity to the NF-κB-IκBα complex; hence, downregulated pNF-κB was observed (Figure 6A,D). Once the free pNF-κB decreased, the number of translocations of this molecule from the cytosol to the nucleus decreased; therefore, proinflammatory cytokine and mediator production diminished in both rFgCyst-treated cells. Nevertheless, in both types of cells, the level of NF-κB was not altered (Figure 6A,D). Previous studies reported different expressions of NF-κB after LPS activation, including downregulation, upregulation, or even remaining constant [64,65,66,67]. These results suggest that the expression of NF-κB induced by LPS depends on the dose and time of incubation. Hence, the correlation between the exact concentration of LPS and incubation time needs further investigation. The proposed model of rFgCyst anti-inflammatory activity is illustrated in Figure 7.

Actually, after activation, macrophages differentiate into classically activated macrophages (M1) and alternatively activated macrophages (M2). M1 plays a crucial role in the immune responses, resulting in the control of infection, but may cause mortality due to sepsis; meanwhile, M2 produces anti-inflammatory cytokines that alleviate excessive inflammation [68,69]. Thus, inducing the polarization of the M1 to M2 phenotype may be an effective treatment to alter uncontrolled inflammation [70,71]. M1 plays a crucial role not only in proinflammatory molecule production but also in antigen-presenting cells (APC). The phagocyted antigens undergo degradation and presentation on the cell surface, associated with MHC-II, CD80, and CD86 as co-stimulatory molecules [20]. Three lysosomal cathepsins (B, L, and S) were evaluated for their role in host immune responses [72]. Cathepsin B is involved in intracellular protein degradation [73]. Cathepsin S and L are important molecules involved in antigen processing with MHC-II formation by degrading the MHC invariant chain (li) in APC [74]. Therefore, the insufficiency of cathepsin S and L results in diminished MHC-II surface expression, thus affecting the immune response [75,76,77]. Meanwhile, M2 usually secretes IL-10, an anti-inflammatory cytokine, for proper Th1 cell and APC regulation [20]. IL-10 performs its effects on APC via the Janus kinase 1 (JAK1)/Tyrosine kinase 2 (Tyk2)/signal transducer and activator of the transcription 3 (STAT3) pathways [78,79]. Parasites may increase the IL-10 levels in the host and decrease the function of M1, thus enhancing the M1 to M2 polarization [80]. 

Previous studies revealed that helminth cystatins were involved in M2 polarization via the modulation of T-cell proliferation, the activation of IL-10 production, and the inhibition of antigen presentation. These activities were illustrated in CPI-2, onchocystatin, and AvCystatin, which are cystatins isolated from *Brugia malayi*, *Onchocerca volvulus*, and *Acanthocheilonema viteae*, respectively [81,82,83]. Furthermore, HCcyst-3, the cystatin derived from *Haemonchus contortus*, also suppressed phagocytosis MHC-II expression and inflammatory cytokine production [84]. Recently, studies have suggested that the recombinant cystatin of *Trichinella spiralis* inhibits the release of TNF-α, IL-1β, and IFN-γ, the expression of iNOS, and induces M2 polarization with the alteration of the antigen processing by downregulating MHC-II in mouse bone marrow-derived macrophages (BMDMs) under the activation of LPS [26,27]. Interestingly, FhCystatin not only downregulated the proinflammatory cytokine but also upregulated anti-inflammatory cytokines including IL-10 [39]. Because of the high identity of type I cystatin between *F. hepatica* and *F. gigantica* [40], we hypothesized that rFgCyst probably upregulates IL-10 as well. Moreover, since rFgCyst revealed its inhibition of mammalian cathepsin L, B, and S [40], we also hypothesized that rFgCyst induces inhibition during antigen processing involved with MHC-II expression; thus, it may lead to M2 polarization promotion. However, these anti-inflammatory aspects require further investigation. However, our results indicate that FgCystatin has a high potential for further development as an anti-inflammatory agent.

## 5. Conclusions

This study primarily demonstrated that rFgCyst exerted anti-inflammatory properties by decreasing the production of proinflammatory cytokines and mediators in LPS-activated macrophage cell lines, including human THP-1-derived macrophages and mouse RAW 264.7 cells through the disturbance of the NF-κB signaling pathway. Our results suggested that rFgCyst affects the production of proinflammatory cytokines (including IL-6, TNF-α, IL-1β, and IL-8) and also mediators (including NO, iNOS, and COX-2) at gene transcription, protein expression, and secretion levels, which ensured that rFgCyst is a potential candidate as an anti-inflammatory agent for the treatment of inflammatory diseases since it affects throughout the production process of proinflammatory molecules. Our findings will be useful as a reference for in vivo studies in both mouse and human models that need to be verified further. 

## Figures and Tables

**Figure 1 pathogens-12-00395-f001:**
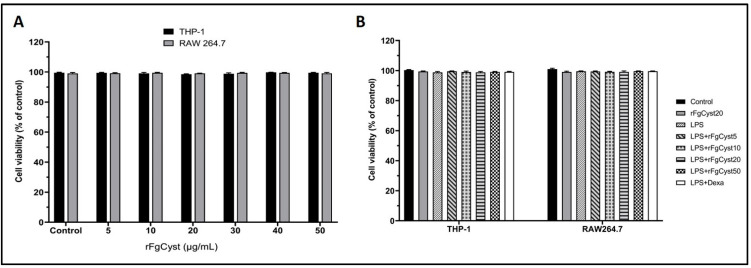
Effect of various concentrations of rFgCyst in THP-1-derived macrophages and RAW264.7 cell viability in the presence (**A**) or absence (**B**) of LPS (500 ng/mL). Five µg/mL of dexamethasone (dexa) was used as a standard anti-inflammatory drug. After 24 h of treatment, the cytotoxicity was determined by MTT assay. The results were evaluated as the percentage of the comparison between the treated cells and the control. The data shown are the mean ± SD from three independent experiments examined in six replicates. One-way ANOVA followed by Dunnett’s multiple comparison test was used for statistical analysis.

**Figure 2 pathogens-12-00395-f002:**
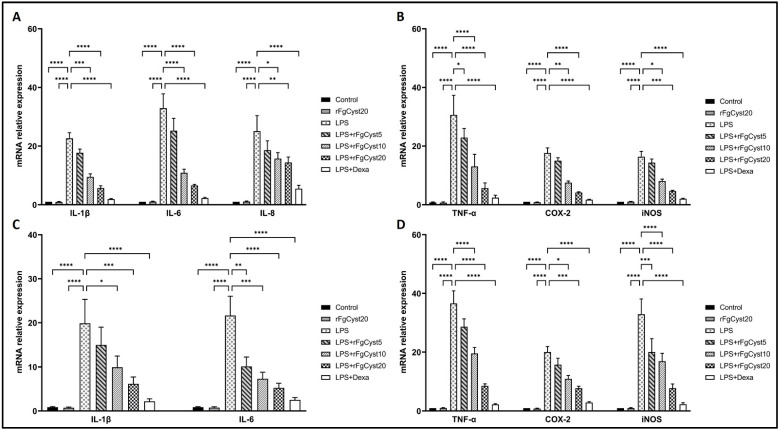
Effects of rFgCyst (5, 10, and 20 µg/mL) on the mRNA expression of proinflammatory cytokines including IL-1β, IL-6, IL-8, and TNF-α and inflammatory mediators including COX-2 and the iNOS mRNA expression in THP-1-derived macrophages (**A**,**B**) and RAW264.7 cells (**C**,**D**) activated with 500 ng/mL LPS. Five µg/mL dexamethasone (dexa) was used as a standard anti-inflammatory drug. After 24 h, the cells were harvested; the gene expression was then determined using qRT-PCR and GAPDH, a housekeeping gene, for normalization. The results of the qRT-PCR of relative molecules are expressed as a relative fold change compared to the control. The data are expressed as the mean ± SD of three independent experiments examined in triplicate. One-way ANOVA followed by Dunnett’s multiple comparison test was used for statistical analysis. * *p* < 0.05, ** *p* < 0.01, *** *p* < 0.001, **** *p* < 0.0001, and ns—nonsignificant represent the level of differences compared to the LPS-stimulated-only control.

**Figure 3 pathogens-12-00395-f003:**
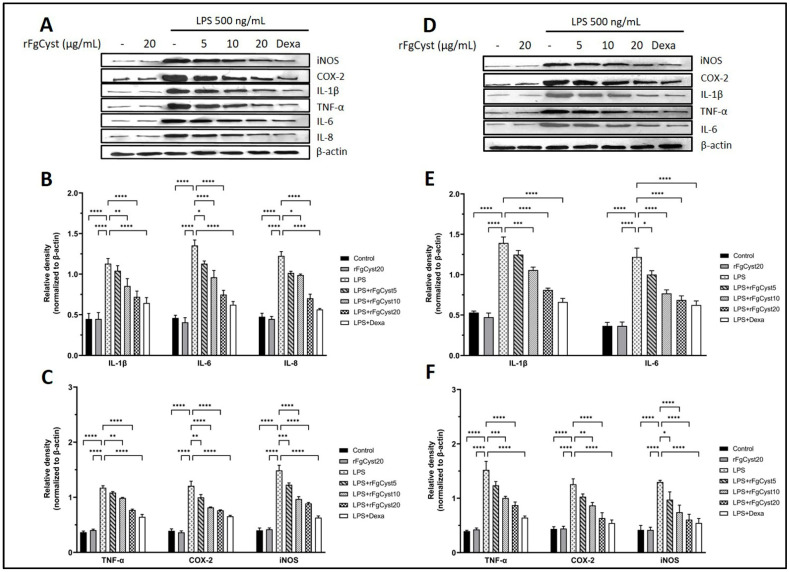
Effects of rFgCyst (5, 10, and 20 µg/mL) on proinflammatory cytokines including IL-1β, IL-6, IL-8, and TNF-α and inflammatory mediators including COX-2 and iNOS protein expression in THP-1-derived macrophages (**A**–**C**) and RAW264.7 cells (**D**–**F**) activated with 500 ng/mL LPS. Five µg/mL dexamethasone (dexa) was used as a standard anti-inflammatory drug. After 24 h, the cells were harvested; the protein expression was then determined using a Western blot. Β-actin was used for normalization as an internal control. The data are expressed as the mean ± SD of three independent experiments examined in triplicate (*n* = 9). One-way ANOVA followed by Dunnett’s multiple comparison test was used for statistical analysis. * *p* < 0.05, ** *p* < 0.01, *** *p* < 0.001, **** *p* < 0.0001, and ns—nonsignificant, represent the level of differences compared to the LPS-stimulated-only control.

**Figure 4 pathogens-12-00395-f004:**
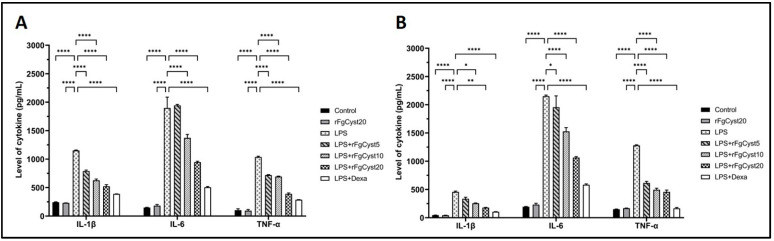
Effects of rFgCyst (5, 10, and 20 µg/mL) on secretory proinflammatory cytokines including IL-1β, IL-6, and TNF-α secretion in THP-1-derived macrophages (**A**) and RAW264.7 cells (**B**) activated with 500 ng/mL LPS. Five µg/mL dexamethasone (dexa) was used as a standard anti-inflammatory drug. After 24 h, the cultured medium was collected, centrifuged, and the level of cytokine secretion was determined using a sandwich ELISA test kit. The data are expressed as the mean ± SD of three independent experiments examined in duplicate (*n* = 6). One-way ANOVA followed by Dunnett’s multiple comparison test was used for statistical analysis. * *p* < 0.05, ** *p* < 0.01, **** *p* < 0.0001, and ns—nonsignificant, represent the level of differences compared to the LPS-stimulated-only control.

**Figure 5 pathogens-12-00395-f005:**
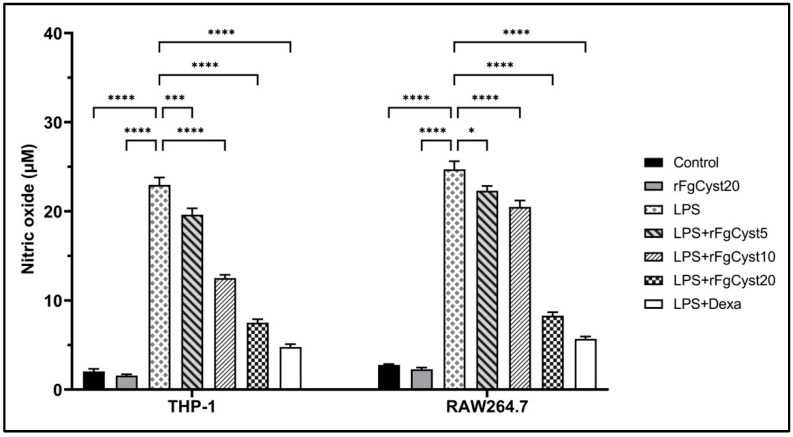
Effects of rFgCyst (5, 10, and 20 µg/mL) on NO production, a proinflammatory mediator in THP-1-derived macrophages and RAW264.7 cells activated with 500 ng/mL LPS. Five µg/mL dexamethasone (dexa) was used as a standard anti-inflammatory drug. After 24 h, the cultured medium was collected and centrifuged, and the level of NO secretion was determined using a Griess assay. The data are expressed as the mean ± SD of three independent experiments examined in six replicates. One-way ANOVA followed by Dunnett’s multiple comparison test was used for statistical analysis. * *p* < 0.05, *** *p* < 0.001, **** *p* < 0.0001 represent differences compared to the LPS-stimulated-only control.

**Figure 6 pathogens-12-00395-f006:**
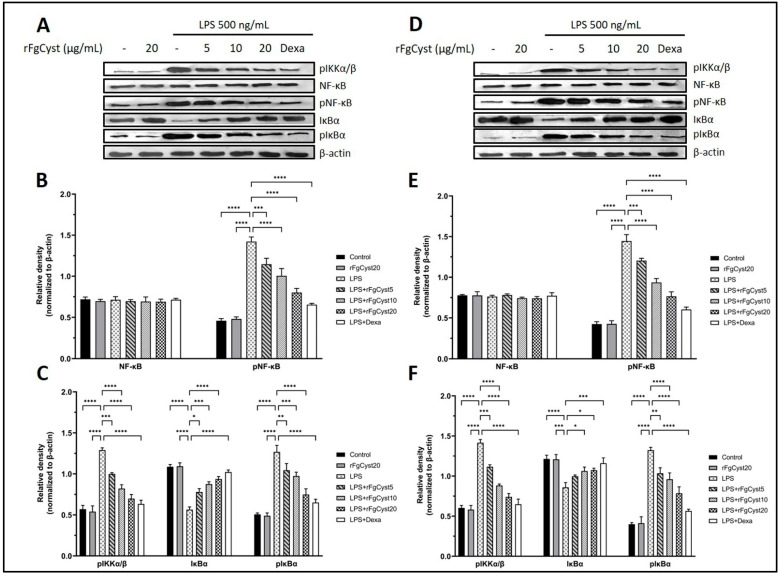
Effects of rFgCyst (5, 10, and 20 µg/mL) on related molecule expression in the NF-κB signaling pathway of THP-1-derived macrophages (**A**–**C**) and RAW264.7 cells (**D**–**F**) activated with 500 ng/mL LPS. Five µg/mL dexamethasone (dexa) was used as a standard anti-inflammatory drug. After 24 h, the cells were harvested; the protein expression was determined using a Western blot. β-actin was used as an internal control for normalization. The data are expressed as the mean ± SD of three independent experiments examined in triplicate (*n* = 9). One-way ANOVA followed by Dunnett’s multiple comparison test was used for statistical analysis. * *p* < 0.05, ** *p* < 0.01, *** *p* < 0.001, **** *p* < 0.0001, and ns—nonsignificant represent the level of differences compared to the LPS-stimulated-only control.

**Figure 7 pathogens-12-00395-f007:**
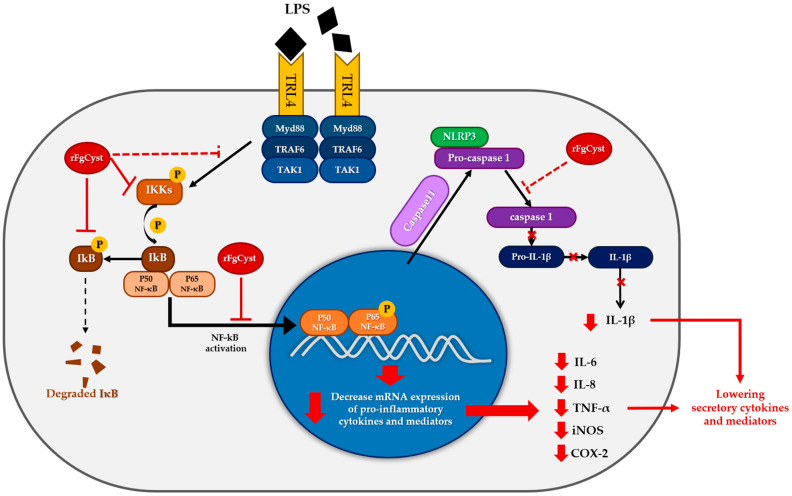
Proposed model for the anti-inflammatory activity of rFgCyst in macrophages. (black lines represent the normal response for LPS activation; red lines represent rFgCyst activity found in this study; and red dashed lines represent the hypothesized activities of rFgCyst).

**Table 1 pathogens-12-00395-t001:** Oligonucleotide primer sequences used for the RT real-time PCR.

Cell	Gene	Accession No.	Primer Sequence from 5′ to 3′		Product Size (bp)
THP-1	IL-1β	NM_000576.3	FW:	CTG AGC TCG CCA GTG AAA TG	202
			RV:	TCC ATG GCC ACA ACA ACT GA	
	IL-6	NM_000600.5	FW:	ACT CAC CTC TTC AGA ACG AAT TG	149
			RV:	CCA TCT TTG GAA GGT TCA GGT TG	
	IL-8	NM_001354840.3	FW:	ACT GAG AGT GAT TGA GAG TGG AC	112
			RV:	AAC CCT CTG CAC CCA GTT TTC	
	TNF-α	NM_000594.4	FW:	GAG GCC AAG CCC TGG TAT G	91
			RV:	CGG GCC GAT TGA TCT CAG C	
	COX-2	NM_000963.40	FW:	CAG CAC TTC ACG CAT CAG TT	128
			RV:	CGC AGT TTA CGC TGT CTA GC	
	iNOS	NM_000625.4	FW:	CGC ATG ACC TTG GTG TTT GG	142
			RV:	CAT AGA CCT TGG GGC TTG CCA	
	GAPDH	NM_001289745.3	FW:	GAG TCA ACG GAT TTG GTC GT	214
			RV:	TGG AAG ATG GTG ATG GGA TT	
RAW 264.7	IL-1β	NM_008361.4	FW:	TGC CAC CCT TTT GAC AGT GAT G	138
			RV:	TGA TGT GCT GCT GCG AGA TT	
	IL-6	NM_031168.2	FW:	CCC CAA TTT CCA ATG CTC TCC	141
			RV:	CGC ACT AGG TTT GCC GAG TA	
	TNF-α	NM_013693.3	FW:	CCC TCA CAC TCA GAT CAT CTT CT	61
			RV:	GCT ACG ACG TGG GCT ACA G	
	COX-2	NM_011198.4	FW:	TGT GAC TGT ACC CGG ACT GG	233
			RV:	TGC ACA TTG TAA GTA GGT GGA C	
	iNOS	NM_001313922.1	FW:	CCC TTC CGA AGT TTC TGG CAG CAG	497
			RV:	GGC TGT CAG AGC CTC GTG GCT TTG G	
	GAPDH	NM_001289726.2	FW:	CGA CTT CAA CAG CGA CAC TCA C	119
			RV:	CCC TGT TGC TGT AGC CAA ATT C	

## Data Availability

Not applicable.

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
