# Peer review of "Type I Cystatin Derived from Fasciola gigantica Suppresses Macrophage-Mediated Inflammatory Responses"

_pathogens, 2023, doi:10.3390/pathogens12030395_

Round 1
Reviewer 1 Report
Dear Editor,
The manuscript presents an important biotechnological advance regarding the application of Fasciola gigantica type-1 cystatins to control the inflammatory response. This potentially lays the foundations for the treatment of patients with autoimmune disorders or difficulties in controlling inflammation, especially in countries where parasitism is minimal and chronic diseases associated with inflammatory processes are more notorious and complex. I highlight the high experience of the authors in describing the methodology and presenting the results, I enjoyed reading the article and it fits perfectly with the approach of the journal and the special section. I congratulate them for the work that has been well-designed, for repeating the experiments, and for every detail that they have taken care of. I am sure it's a big step toward something brighter: a clinical trial.
I have made comments that I hope will be constructive and serve to clarify or tidy up the manuscript a bit more, which may be accepted after a minor revision.
Best wishes.
Abstract
- I suggest adding something slight regarding the main methodologies employed and indicating that the study will be in a human cell lineage.
- In lines 26 and 27, the wording could be improved to indicate that IL-1β, IL-6, and TNF-α secretion, in addition to NO production, were decreased.
- Perhaps in the keywords they could replace the words that already appear in the title and thus enrich the indexing.
Introduction
- In the second sentence (lines 36 and 37), consistent with the above, one could elaborate a bit more on the first noxa-dependent activated immunogenic and antigenic mechanisms. I suggest that the first three citations be different, such as some book, book chapter, or review article on what was discussed. Primary articles are cited in conceptual preambles.
- Avoid the reiterative use of the word "excessive" in the first paragraph, as they are in close proximity.
- Perhaps the sentence that begins in line 39 should be moved to the end of the same paragraph, it would read better.
- In line 53, I suggest strengthening what is being said about the helminth-habitat association and its relationship to inflammatory modulation. Cite as much as possible.
- What is stated in line 56 sounds like a hasty conclusion. In any case, change to future tense.
- In the paragraph beginning on line 58, I recommend adding that different cystatins are produced in almost all nucleated cells of humans and helminths.
- I think the sentence on line 71 should be deleted as it overlaps with line 58.
- Employ a connector to start the paragraph located at line 85. Also, the first sentence could be enhanced by adding interesting facts such as the current economic loss in animal husbandry and the number of people with fasciolosis around the world. Further on, remember that, in humans, Fasciola migration also frequently occurs to other organs such as the lung and even the eyes.
Materials and Methods
- In method 2.1, did you perform an evaluation of the quantity and quality of the cDNA obtained prior to cloning into the pGEM-T vector? Personally, seeing that you then performed sequencing, I would like to know what average or minimum concentration of nucleic acids you worked with at this stage. Here I would also like to know where the Fasciola gigantica specimens you worked with in your research came from: are they commercial strains, wild, or from a biorepository of your institution?
References
- Check the instructions for authors and homogenize the structure of the doi, in some cases, they appear with "https" and in others not.
- Similarly, homogenize the names of the journals, in some cases, they appear abbreviated, and in others with full names. Standardize based on the requirements of the journal.

Reviewer 2 Report
This manuscript from Chantree et al reports the anti-inflammatory properties of a type-1 cystatin from the trematode Fasciola gigantica.
The findings in this report align with the findings of other anti-inflammatory activity for cystatins from other helminths.
The data appears sound and supports the conclusions.
I only have minor suggestions for changes to the manuscript, including the following:
1) Where the authors cite different cystatins in other parasites in the introduction and discussion, it would be useful for comparative purposed if the authors stated the "type" of the cystating (e.g. type-1, type-2, etc).
2) At line 68 the authors state type-3 cystatins are synthesized in the liver. The following statement says cystatin-3s can be found in humans and helminths - where are they in helminths if synthesized in the liver in humans? If known, add a sentence on their typical localisation in helminths.
3) line 86: delete the word "animal".
4) Suggest re-wording line 561-562. It currently reads like the authors are discussing transcriptional and translational levels of rFgCyst. Suggest: "The anti-inflammatory effects of rFgCyst on the transcription and translation of immune-mediating proteins in THP-1 derived macrophages or RAW264.7 cells was evaluated".
Reviewer 3 Report
The Msc deals with the anti-inflammation induced by the recombinant type-1 cystatin from Fasciola gigantica (rFgCyst). All the work was done in vitro.
The work is well presented and and the methods well described. Statistical analyses were performed and seem relevant.
Nevertheless, I think that most of the work submitted was already done with the homologous cytatin from Fasciola hepatica(rFhCyst) (as introduced by author iself). The only originality is on the NF-KB pathway analyze since the results on cytokines are already known for rFhCyst.
About the Msc, it is well written excepted the first third of the discussion which for me is a repetition of the introduction section.
Round 2
Reviewer 3 Report
I agree with the author answer to the main question I adressed in the previous report.
I think that there are a lot of things in the article and it is difficult to see all the messages mainly on these novelties. I suggest to authors to point it in the conclusion to help the readers.
Author Response
Dear reviewer,
Thank you very much for your suggestion. The conclusion has been rewritten, and the novelties of our study have been added to this part.
Kind regards